# Acute or Chronic Exposure to Corticosterone Promotes Wakefulness in Mice

**DOI:** 10.3390/brainsci13101472

**Published:** 2023-10-18

**Authors:** Zhen Yao, Bei-Xuan Zhang, Hui Chen, Xiao-Wei Jiang, Wei-Min Qu, Zhi-Li Huang

**Affiliations:** Department of Pharmacology, School of Basic Medical Sciences, State Key Laboratory of Medical Neurobiology and MOE Frontiers Center for Brain Science, Institutes of Brain Science, Fudan University, Shanghai 200032, China; 18111010019@fudan.edu.cn (Z.Y.); 22111520066@m.fudan.edu.cn (B.-X.Z.); 19111520042@fudan.edu.cn (H.C.); catherine_jxw@outlook.com (X.-W.J.); quweimin@fudan.edu.cn (W.-M.Q.)

**Keywords:** corticosterone, sleep, wakefulness, depression, stress, insomnia

## Abstract

Elevated glucocorticoid levels triggered by stress potentially contribute to sleep disturbances in stress-induced depression. However, sleep changes in response to elevated corticosterone (CORT), the major glucocorticoid in rodents, remain unclear. Here, we investigated the effects of acute or chronic CORT administration on sleep using electroencephalogram (EEG) and electromyography (EMG) recordings in freely moving mice. Acute CORT exposure rapidly promoted wakefulness, marked by increased episodes and enhanced EEG delta power, while simultaneously suppressing rapid eye movement (REM) and non-rapid eye movement (NREM) sleep, with the latter marked by decreased mean duration and reduced delta power. Prolonged 28-day CORT exposure led to excessive wakefulness and REM sleep, characterized by higher episodes, and decreased NREM sleep, characterized by higher episodes and reduced mean duration. EEG theta activity during REM sleep and delta activity during NREM sleep were attenuated following 28-day CORT exposure. These effects persisted, except for REM sleep amounts, even 7 days after the drug withdrawal. Elevated plasma CORT levels and depressive phenotypes were identified and correlated with observed sleep changes during and after administration. Fos expression significantly increased in the lateral habenula, lateral hypothalamus, and ventral tegmental area following acute or chronic CORT treatment. Our findings demonstrate that CORT exposure enhanced wakefulness, suppressed and fragmented NREM sleep, and altered EEG activity across all stages. This study illuminates sleep alterations during short or extended periods of heightened CORT levels in mice, providing a neural link connecting insomnia and depression.

## 1. Introduction

Sleep, characterized by distinct stages including non-rapid eye movement (NREM) and rapid eye movement (REM) sleep, is a fundamental physiological process essential for survival and optimal functioning [1]. Sleep disturbances represent one of the core symptoms of various psychiatric disorders [2], including depression [3]. Abnormalities in sleep architecture are frequently observed in individuals with depression, which manifests as insomnia or hypersomnia [4]. Common sleep characteristics in depressed patients also include reduced slow-wave sleep and increased sleep discontinuity [5]. Conversely, sleep disorders can act as a major trigger for depression. Meta-analyses of longitudinal studies have established insomnia as a predictor of depression onset [6], while other investigations have demonstrated that any form of sleep disorder constitutes a risk factor for developing depression [7]. Improving sleep quality, on the other hand, contributes positively to overall mental well-being [8]. Gaining an understanding of alterations in sleep architecture during the progression of depression may provide insights for intervention strategies.

Stress, as a response to novel or threatening stimuli, constitutes a main risk factor in the development of depression [9]. Acute or chronic stress has been found to induce anxiety and depression in humans [10], or depressive-like behaviour in rodents [11]. Recent research has suggested the intricate associations between stress and sleep. Stress induced insomnia/hyperarousal in humans and rodent models, and the underlying neural circuits were dissected [12,13]. However, various forms of stress have also been found to increase the amounts of REM and NREM sleep [14,15,16]. The exposure to acute or chronic stress prompts a consequently heightened secretion of glucocorticoid hormones, such as cortisol in humans and corticosterone (CORT) in rodents, despite the presence of negative feedback [17]. Elevated cortisol levels have been purposed to underlie the pathogenesis of depression and anti-cortisol medications have demonstrated beneficial results in depressed patients [18,19]. In rodent models, the acute or chronic administration of CORT mimics the physiological states experienced by humans exposed to stressful conditions, of whom blood glucocorticoid levels increase rapidly or gradually [20]. A few studies have attempted to explore the effects of CORT exposure on sleep architecture. For instance, single injections of CORT in rats were found to produce an initial enhancement of wakefulness and a reduction in slow-wave sleep within the first hour [21], while repeated injections over 7 days led to a decrease in total NREM sleep [22]. Further, 21 days of CORT treatment reportedly shortened REM sleep latency [23]. However, contradictory findings have emerged from a study conducted in mice, which reported that after 4 weeks of CORT treatment, NREM sleep significantly increased [24]. These divergent outcomes have primarily focused on the short-term (7-day) effects of CORT on sleep/wake patterns or exhibited limitations regarding the depth of sleep quality analysis. Moreover, investigations into alterations in sleep/wake cycles post-drug withdrawal have remained scarce. Nevertheless, the roles and potential regulatory mechanisms of acute and chronic elevation in CORT levels in sleep disturbances have not been clarified.

In this study, we conducted a comprehensive analysis of sleep/wake parameters following acute and chronic (28-day) exposure to CORT, extending to a 7-day post-treatment cessation phase. The acute CORT exposure paradigm has previously been reported to elevate plasma CORT levels, with peak concentrations observed 1–4 h after administration [25]. Therefore, we selected the timepoint for administering exogenous CORT when the normal physiological CORT level was low, but gradually increasing and reaching its peak several hours later. This specific timepoint also coincided with a high spontaneous sleep drive. Our aim was to identify the implications of raised glucocorticoid levels in sleep disturbances. We also assessed concurrent behavioral phenotypes and dynamic profiles of plasma CORT during and after the chronic exposure. Immunofluorescence staining was performed to delve into potential neural activities associated with acute and chronic CORT-induced behaviour abnormalities. By delineating the effects of elevated glucocorticoids on sleep architecture and investigating potential mechanisms, our study aims to contribute to understanding of the intricate interplay between depression and sleep disorders.

## 2. Materials and Methods

### 2.1. Animals

Adult male C57BL/6J mice (8–14 weeks, 22–26 g) were obtained from Shanghai Silaike Experiment Animal Company. Mice were housed at an ambient temperature of 22 ± 0.5 °C, with a relative humidity of 60% ± 2%. A 12 h light/dark cycle (100 Lux, lights on at 07:00 a.m.) was automatically controlled [26]. Food and water were available ad libitum. Two randomly assigned cohorts of animals were used, respectively, in the plasma CORT measurement and electroencephalogram (EEG)/electromyography (EMG) recordings, to avoid the influence of surgery. Two other cohorts were used in the rest of the behavioral testing experiments: one cohort in (by the testing order) the sucrose preference test (SPT), open field (OF) test, self-grooming test, elevated plus maze (EPM) test, and tail suspension test (TST); and the other cohort in the novelty-suppressed feeding (NSF) test. An interval of 30 minutes (min) was set between the two behavioral testing experiments. All animal experiments were approved by the Medical Experimental Animal Administrative Committee of Shanghai. All experimental procedures were approved by the Committee on the Ethics of Animal Experiments of the School of Basic Medical Sciences, Fudan University (license identification number: 20200306-023). During all of the experiments, we tried our best to minimize the pain and discomfort of animals.

### 2.2. CORT Exposure

During the acute administration procedure, CORT (20 mg/kg, MedChemExpress, Monmouth Junction, NJ, USA), suspended in saline with 1% Tween 80 and 0.1% DMSO, was injected subcutaneously at Zeitgeber time (ZT) 1, with vehicle injected 1 week earlier to the same animal at ZT 1 as a vehicle control (in the c-Fos staining experiment, the vehicle group was composed of the littermates of the CORT group). During the chronic administration procedure (from Day 1 (D1) to D28), CORT was given via drinking water by being dissolved in double-distilled water, with the help of beta-cyclodextrin and ultrasound, to reach a final concentration of 0.035 mg/mL. Vehicle-administered mice only received beta-cyclodextrin in the drinking water (4.5 mg/mL) without CORT [27]. These doses were chosen based on the series work of Dieterich, A and his colleagues, focusing on the effects of CORT administration in mice [28].

### 2.3. Plasma CORT Measurements

To confirm the circulating CORT levels, blood samples were taken from mice at ZT 7–8 on D1, D14, and D28 of the chronic administration course, and 1 day (D29) and 7 days (D35) after the drug withdrawal. Immediately after opening the cage, mice were anesthetized and blood was collected from the heart and placed into a heparin sodium tube (Huabo Medical Care, Shandong, China). Plasma was carefully extracted after centrifugation at 4 °C. Plasma CORT levels were measured using an enzyme-linked immunosorbent assay (ELISA) kit (EM30880S, Biotech WELL, Changzhou, China) according to the manufacturer’s recommended protocol.

### 2.4. Surgery

According to previous studies [29,30,31], all mice were anesthetized with chloral hydrate (360 mg/kg, intraperitoneal injection) for surgical procedures and were placed in a stereotaxic apparatus (RWD, Shenzhen, China). Mice were implanted with electrodes for EEG and EMG recordings. Two stainless steel screws of EEG electrodes were inserted into the exposed skull after the skin was cut open at 1.5 mm from the midline, 1.0 mm anterior to the bregma, and 3.0 mm posterior to the bregma. Two EMG cables were inserted into the trapezius muscle. Finally, the electrodes were immobilized using dental cement and attached firmly to the skull. The scalp wound was sutured, and the mice were then placed on a heating pad until it resumed normal activity.

### 2.5. EEG/EMG Recordings and Analysis

After at least 1 week of recovery from the EEG/EMG electrodes’ implantation, the mice were transferred to the recording room and housed individually in transparent barrels for habituation. The electrodes of each mouse were linked to a cable that was connected to a slip ring. The mice were habituated to the recording cables and conditions for at least 3 days before each recording, after which mice were transferred back to their home cage.

Cortical EEG and EMG signals were amplified and filtered (EEG, 0.5–30 Hz; EMG, 20–200 Hz) at a sampling rate of 128 Hz and recorded using VitalRecorder (Kissei Comtec, Nagano, Japan). When complete, the recordings were automatically scored offline every 4 s as wakefulness, NREM sleep, or REM sleep using SleepSign 3.0, based on the published standard criteria [32,33]. Defined sleep/wake stages were manually corrected, if necessary. The investigators who checked the EEG signals to determine the brain states were blinded to the group information.

### 2.6. OF Test

In a quiet environment, mice were gently placed in the center of the bottom surface of an open field chamber (50 cm × 50 cm × 40 cm). Mice were allowed to move freely for 10 min in the chamber and after each trial the chamber was cleansed with 75% ethanol. The bottom plane of the open field chamber was divided into 25 squares, with the central 9 squares forming the center zone. The time and trace that the mice travelled through the center zone and total area were recorded by a high-definition camera that was placed above the chamber and analyzed using the connected video-tracking system (The Tracking Master V3.0, TMV3).

### 2.7. SPT

Two bottles of double-distilled water were placed in the cage in advance for the mice to acclimatize. The day before the experiment day, all distilled water was replaced with 1% (*w*/*v*) sucrose water. On the day of the experiment, one of the sucrose water bottles was replaced with a distilled water bottle. The positions of the bottles were switched over the 24 h period to avoid place preference of mice. Sucrose preference is the ratio of sucrose water consumption relative to the total consumption of both sucrose and distilled water during the experiment day, as measured by the changes in weight.

### 2.8. EPM

The EPM consists of a pair of opposite open arms and a pair of opposite closed arms and a central region, which cross each other in a cross shape. At the beginning of the experiment, each mouse was placed in the central region, while the videography and recording were switched on, and allowed to explore freely for 5 min. The video and TMV3 system were used to analyze the time the mice spent in the open arms and the trace that the mice travelled within the experimental window.

### 2.9. TST

Each mouse was placed in a chamber, which was closed on 3 sides and open on 1 side, and was suspended by its tail with its head hanging 30 cm from the ground using a piece of tape that was strong enough to prevent the mouse from falling without damaging the skin of the tail. Once immobilized, the mouse was unable to escape or grasp nearby surfaces (a mouse was discarded if it climbed onto the tape by itself). The test lasted 6 min and the immobility time of the last 5 min was measured by the video and TMV3 system.

### 2.10. NSF Test

The mice were fasted for 24 h (while water was provided ad libitum). At the beginning of the experiment, each mouse was placed in a new cage with a weighed dry food pellet on a piece of white filter paper positioned in the center of the cage. The latency of mice to first bite the food was monitored within the 5 min experimental window.

### 2.11. Self-Grooming Counts

After the mice had been adapted in advance, they were placed in the experimental area, and a camera tracked and recorded their behaviour, recording the number of spontaneous grooming within half an hour. The method of defining a complete grooming behaviour is referred to in this literature [34].

### 2.12. Immunohistochemistry

Mice were deeply anesthetized and transcardially perfused with 0.1 M cold phosphate-buffered saline (PBS), followed by 4% paraformaldehyde (PFA) in PBS. The brains were removed and postfixed overnight in 4% PFA at 4 °C, and then incubated in 20% sucrose in 4% PFA and 30% sucrose in PBS for 24 h, respectively. After been embedded in OCT compound, each brain sample was sliced into 30 µm coronal sections using a freezing microtome (CM1950, Leica, Wetzlar, Germany).

For fluorescent detection of c-Fos [35], the free-floating brain slices were washed 3 times with PBS and incubated with primary rabbit antibody (anti-c-Fos, ABE457, Millipore, Burlington, MA, USA) diluted in PBST (0.3% Triton X-100 in 0.1 M PBS, 1:10,000) for 48 h at 4 °C. Slices were then washed with PBS and incubated with donkey anti-rabbit Alexa Fluor-conjugated secondary antibody (1:1000, Jackson ImmunoResearch, West Grove, PA, USA) for 2 h at room temperature. Slices were washed in PBS and mounted onto microscope slides with Fluoromout-G^TM^ (Southern Biotech, Birmingham, AL, USA).

### 2.13. Cell Counting

After immunostaining experiments, images of the brain slices were captured using a high-resolution fluorescence microscope (Olympus VS120 slide scanner, Tokyo, Japan). The boundary of the target brain regions was delineated by overlapping the brain map of mouse (Paxinos and Franklin, 2013) onto the images of the coronal slices with the help of markers such as the third ventricle (3v). Neurons showing positive c-Fos immunoreactivity in the range of target brain regions were counted using ImageJ software (https://imagej.net/ij/download.html).

### 2.14. Statistical Analysis

All data are presented as the mean ± standard error of the mean (SEM). Comparisons of time course changes in sleep/wake stages and other behavioral phenotypes of vehicle- and CORT-administered mice were performed using two-way ANOVA with Geisser–Greenhouse correction followed by a Bonferroni post hoc test. Comparisons of plasma CORT levels, total sleep-wake amounts, episode number, mean duration, power density, and c-Fos-positive cell number between the CORT and the vehicle group were analyzed by an unpaired, two-tailed Student’s *t*-test. Sleep/wake parameters and other behavioral results among CORT groups on different experiment days were evaluated using a paired, two-tailed Student’s *t*-test. Prism 8.0.2 (GraphPad Software) was used for all statistical analyses. In all cases, *p* < 0.05 was considered significant.

## 3. Results

### 3.1. Acute CORT Administration Induced a Rapid Increase in Wakefulness and a Decrease in REM and NREM Sleep Accompanied by the Altered EEG Delta Power

We first accessed the effect of an acute rise in CORT levels on the sleep/wake profiles in mice. The vehicle or CORT was administered by subcutaneous injection at ZT 1, when mice typically exhibit high spontaneous sleep drive. The EEG/EMG was recorded simultaneously for 24 h starting at ZT 0.

Following CORT injection, a significant increase in wakefulness was elicited during ZT 2–3 (*p* = 0.0053) and ZT 4–5 (*p* = 0.0444), accompanied by a compensatory decrease during ZT 14–15 (*p* = 0.0183) compared to the vehicle control. A concomitant reduction in NREM sleep was observed during ZT 2–3 (*p* = 0.0059) and ZT 4–5 (*p* = 0.0268), with a compensatory increase during ZT 14–15 (*p* = 0.0374), and REM sleep also experienced a significant reduction during ZT 1–2 (*p* = 0.0127) and ZT 2–3 (*p* = 0.0460), relative to the vehicle control (Figure 1A,B). Over the 4 h period following CORT injection, there was a significant increase in total wakefulness (*p* = 0.0007), accompanied by decreased amounts of REM sleep (*p* = 0.0404) and NREM sleep (*p* = 0.0009) (Figure 1C). Key indicators for assessing sleep/wake behaviour, such as episode number, mean duration, and EEG power density reflecting sleep architecture and quality, were also examined [32,33]. Episodes of wakefulness displayed a significant increase during the 4 h following CORT injection (*p* = 0.0340), accounting for the added wakefulness during this period, while the decreased NREM sleep during these 4 h primarily resulted from the significantly reduced mean duration (*p* = 0.0011) (Figure 1D). During wakefulness within these 4 h, a significant increase in EEG power within the delta band (0.5–4.0 Hz) was observed after CORT injection. In contrast, NREM sleep displayed significantly lower EEG delta power after CORT injection compared to control (Figure 1E). To sum up, an acute surge in CORT levels was found to rapidly increase wakefulness in mice with noticeable rebounds later that day, as well as to enhance EEG delta activity during wakefulness and attenuate EEG delta activity during the dampened NREM sleep.

### 3.2. Chronic CORT Administration Elevated CORT Levels and Induced Depressive-like Behaviour in Mice

Prior to exploring the effect of prolonged exposure to high glucocorticoid levels on sleep/wake cycles, we investigated plasma CORT dynamics and behavioral phenotypes in mice. To avoid potential adverse skin reactions and chronic stress arising from repeated subcutaneous injections [36,37], CORT was administered through drinking water. The experimental timeline is shown in Figure 2A.

Plasma CORT levels displayed no significant alterations during the initial phase of chronic exposure (D1 to D14), remaining within the normal physiological range, since the blood samples were taken around the time at which circulating CORT levels peak in mice [38]. However, following 28 days of administration, the CORT group exhibited a significant elevation by 459.64% (*p* = 0.0019) relative to the vehicle control. There was no significant difference in the fluid consumptions between the CORT-treated mice and vehicle-treated mice (Figure 2C) across the administration course, suggesting that the increase in circulating CORT levels was not due to the increased intake of CORT-containing water. This elevation retained statistical significance during the following days tested after the drug withdrawal (D29, *p* = 0.0007, and D35, *p* = 0.0456) (Figure 2B), indicating that aberrantly elevated circulating CORT levels resulting from long-term CORT administration persisted at least 7 days after the drug withdrawal. 

Chronic CORT exposure reduced body weight gain and induced depressive-like phenotypes in mice (Figure 2D–J). These effects became significant on D28 of administration, while remaining inconspicuous within the initial 2-week period. To elaborate, on D28, the body weight of CORT-treated mice was significantly lower than the control mice (*p* = 0.0185, Figure 2D). Moreover, compared to vehicle control, mice showed a significantly decreased ratio of time spent in the center zone relative to the total area during the OF test (*p* = 0.0359, Figure 2E) and a significant elongation in feeding latency during the NSF test (*p* = 0.0487, Figure 2G), both indicative of depressive states in animals [39,40]. Mice also exhibited a significantly decreased sucrose preference during the SPT (*p* = 0.0036, Figure 2F), a measure for the anhedonia-like state, and increased immobility duration during the TST (*p* = 0.0060, Figure 2H), a measure for the despair-like state. Aligned with the CORT dynamics, the poor body weight gain and these depressive-like phenotypes were maintained 7 days after the drug withdrawal. Noticeably, no significant difference during the spontaneous grooming bouts (Figure 2I) and the EPM test (Figure 2J) were found between the CORT and vehicle groups throughout the administration and subsequent periods; both methods were employed to access anxiety-like behaviour in animals. Taken together, these results show that mice exhibited depressive-like behaviour after 28-day CORT administration.

### 3.3. Chronic CORT Administration Promoted Wakefulness and REM Sleep While It Suppressed NREM Sleep

Next, we investigated the effects of chronic CORT exposure on sleep/wake patterns in mice throughout the 28-day administration and the subsequent drug withdrawal period. The experiment timeline is shown in Figure 3A.

During the initial 2 weeks, no notable alterations in wakefulness or NREM sleep emerged. However, after continuous 28-day CORT administration, mice displayed substantial increases in wakefulness in three distinct 2 h timeframes (ZT 0–2, *p* = 0.0322; ZT 4–6, *p* = 0.0292; and ZT 8–10, *p* = 0.0093) during the light phase and concomitant reductions in NREM sleep (ZT 0–2, *p* = 0.0297; ZT 4–6, *p* = 0.0404; and ZT 8–10, *p* = 0.0096) relative to the vehicle control. Noticeably, the changes in wakefulness and NREM sleep patterns remained pronounced even 1 week following drug withdrawal, with a 2 h timeframe (ZT 8–10) reflecting heightened wakefulness (*p* = 0.0323) and concomitantly decreased NREM sleep (*p* = 0.0476) compared to the vehicle group (Figure 3B,D). 

The change pattern in REM sleep differed from those observed in the other two stages. There was an increase in REM sleep in a 2 h timeframe after 14-day CORT administration during the light phase (ZT 6–8, *p* = 0.0300), and this increase persisted over the subsequent 14 days but in different timeframes (ZT 2–4, *p* = 0.0319; ZT 10–12, *p* = 0.0349). In contrast to the effects on wakefulness and NREM sleep, REM sleep amounts returned to control levels 1 week after the termination of CORT exposure (Figure 3C).

In brief, CORT exposure engendered time-dependent enhancements in wakefulness and REM sleep, coupled with reductions in NREM sleep over the 28-day CORT exposure period. The effects on wakefulness and NREM sleep were maintained for at least 7 days following the drug withdrawal.

### 3.4. Changes in Wakefulness, REM Sleep, NREM Sleep, and REM Sleep Ratio Induced by Chronic CORT Exposure Predominantly Occurred during the Light Phase

We proceeded to compare the total amounts of time spent in the wake, REM sleep, and NREM sleep stages, and the REM sleep ratio during 24 h, the light phase, and the dark phase, throughout and following the chronic CORT exposure regimen. The increase in wake amounts induced by CORT administration during an entire 24 h cycle reached statistical significance on D28 (*p* = 0.0080) and persisted until D35 (*p* = 0.0052), 7 days following drug withdrawal, in comparison to vehicle control or the CORT group on D1 (D28, *p* = 0.0300; D35, *p* = 0.0430) (Figure 4A). Concomitantly, total NREM sleep duration was significantly decreased on D28 and D35 relative to the vehicle control (D28, *p* = 0.0020; D35, *p* = 0.0056) or the D1 level of CORT-exposed mice (D28, *p* = 0.0053; D35, *p* = 0.0477) (Figure 4C). 

Further dividing the 24 h cycle into 12 h light and dark phases, a significant increase in the amount of wakefulness (16.79%, *p* = 0.0032) was discernible after 28 days of CORT administration, and this increase was maintained 7 days later (10.38%, *p* = 0.0225) during the light or “resting” phase, in comparison to the vehicle control. In parallel, CORT exposure markedly attenuated NREM sleep amounts during the light phase, resulting in a 15.76% decrease on D28 (*p* = 0.0021) and a 10.13% decrease on D35 (*p* = 0.0150) relative to the vehicle control (Figure 4A,C). In contrast, no notable alterations in sleep/wakefulness durations were observed during the dark or “active” phase. 

For REM sleep, the total amount across the 24 h cycle showed a trend to increase on D14 and D28 of CORT exposure, although these changes did not reach significance. However, during the light phase on D14 and D28, a significant increase in REM sleep amounts was observed in the CORT group compared to the vehicle control (D14, 13.27%, *p* = 0.0300; D28, 13.44%, *p* = 0.0436) (Figure 4B). Given that REM sleep exhibited an increase despite the reduction in total sleep duration, we analyzed the ratio of REM sleep to total sleep amounts throughout and following chronic administration. This analysis revealed an increase in REM sleep ratio during the latter half of the administration phase (D14 and D28) compared to the vehicle group. Moreover, during the light phase, the REM sleep ratio was increased significantly across all weeks tested during and 7 days after the 28-day CORT exposure, in comparison with the vehicle group or the CORT group on D1 (Figure 4D). 

Collectively, these findings suggest that effects of chronic exposure to elevated glucocorticoid levels on sleep/wake parameters are primarily manifested during the inactive phase.

### 3.5. Chronic CORT Administration Increased Episodes of Wakefulness, REM Sleep, and NREM Sleep, Prolonged REM Sleep Duration and Shortened NREM Sleep Duration

Here, the total number of episodes and mean duration of each stage during the 12 h light and 12 h dark phases were analyzed to measure the fragmentation level and stability of sleep/wake cycles. 

Our analysis revealed that the episodes of wakefulness were significantly increased on D28 of CORT exposure (*p* = 0.0402) and persisted 7 days later (*p* = 0.0479) during the light phase (Figure 5A), which accounted for the observed increase in wakefulness on these two days (Figure 4A). Interestingly, during the light phase, 28 days of CORT administration led to a significant increase in REM sleep episodes (*p* = 0.0449), while 14 days showed a trend to an increase (*p* = 0.1460) (Figure 5A), contributing to the increase in REM sleep during these two light phases (Figure 4B). By contrast, the mean duration of REM sleep was significantly prolonged during the dark phase on both D14 (*p* = 0.0231) and D28 (*p* = 0.0465) of CORT administration (Figure 5D). NREM sleep, on the other hand, exhibited a significant increase in episode number on the last day of CORT administration during the light phase (*p* = 0.0362) (Figure 5A), despite a dramatic reduction in the total NREM sleep amounts on this day (Figure 4C). The mean duration of NREM sleep displayed a remarkable decrease on D14 (*p* = 0.0377) and D28 (*p* = 0.0009), relative to the vehicle control, during the light phase (Figure 5B), most likely contributing to the reduced NREM sleep during this phase on D28 (Figure 4C). This reduction in mean duration during the dark phase reached significance on D28 as well (*p* = 0.0464) (Figure 5D). 

In summary, chronic CORT exposure led to an increased number of episodes in wakefulness, REM sleep, and NREM sleep, with REM sleep showing a prolonged mean duration and NREM sleep showing a shortened one. These findings suggest that the promotion of wakefulness and REM sleep was primarily driven by increased episode occurrences, and the suppression of NREM sleep is associated with a higher level of fragmentation following chronic CORT exposure.

### 3.6. Chronic CORT Administration Decreased EEG Delta Activity and Increased Theta Activity during Wakefulness, While It Decreased EEG Theta Activity and Delta Activity during REM and NREM Sleep, Respectively

To examine the effects of chronically elevated CORT levels on sleep quality, EEG power spectra were analyzed. For the wake state, during the light phase, there was a significant decrease in EEG power within the delta band across all days tested post-D1, contrasting with increased EEG power within the theta range (6–10 Hz) found on D28 of CORT administration. EEG power within the alpha range (12–14 Hz) was decreased across chronic administration (D1, D14 and D28). By contrast, changes during the dark phase were less pronounced, with delta power decreased on D28 and D35, theta power increased on D35, and alpha power increased on D1 and D35, relative to the vehicle control (Figure 6A). Nevertheless, alterations in wake EEG power density point towards heightened attentiveness and alertness in mice following chronic CORT exposure.

For the REM sleep state, a significant decrease in theta power during the light phase was detected on D14 of CORT administration and this reduction was even more pronounced on D28 and D35, compared to the vehicle control. During the dark phase, theta power consistently decreased throughout the post-D1 weeks, exhibiting the strongest effect on D35, in contrast to no change in REM sleep amounts being observed after the CORT treatment ceased (Figure 4B). There was also an increase in alpha power on D14 and D28 of administration (Figure 6B). 

For the NREM sleep state, during both light and dark phases, delta power was decreased across all days tested relative to the vehicle control. This attenuation was strongest on D28, suggesting a correlation between reduction in delta power and the duration of CORT exposure, persisting significantly until D35. Moreover, in the light phase, a dose-dependent decrease in theta power was observed from D1 to D28 of CORT exposure, and this reduction also remained significant at least 7 days after the drug withdrawal (Figure 6C).

### 3.7. Effects of Acute and Chronic CORT Administration on c-Fos Protein Expression in Certain Brain Regions

We have demonstrated the effects of acutely or chronically elevated CORT levels on sleep architecture, mainly characterized by increased wakefulness, and the induction of depressive phenotypes following chronic CORT exposure. To further elucidate the potential mechanisms underlying CORT-induced wakefulness and depression, we performed c-Fos immunostaining, a transcription factor widely used as a marker of neuronal activity [41], in the brains of mice after acute CORT injections at ZT 1 or chronic CORT administration via drinking water over a 4-week span. The control group received vehicle treatment within the same timeframes (Figure 7A). Figure 7B,D illustrate that a few neurons expressed c-Fos in the lateral habenula (LHb), lateral hypothalamus (LH), and ventral tegmental area (VTA) following vehicle injections, whereas a significant increase in c-Fos expression in the LHb (*p* = 0.0358), LH (*p* < 0.0001), and VTA (*p* = 0.0360) was found in acute CORT-exposed mice. An increase in the number of c-Fos-positive cells in the basolateral amygdala (BLA) was also observed, although the enhancement was not significant (*p* = 0.0565). A different activation pattern was observed in the paraventricular nucleus of the hypothalamus (PVH), where acute CORT exposure induced a c-Fos reduction trend without reaching statistical significance (*p* = 0.4182). In contrast, more robust expression of c-Fos was observed in the LHb, PVH, BLA, LH, and VTA in mice receiving prolonged CORT treatment, compared to the vehicle group (Figure 7C). The quantitative analysis revealed a significant increase in the LHb (*p* = 0.0481), PVH (*p* = 0.0241), BLA (*p*  =  0.0002), LH (*p* = 0.0003), and VTA (*p* = 0.0003) of chronic CORT-exposed mice relative to the vehicle control (Figure 7E). These findings likely underlie the shared and distinct effects of acute and chronic CORT exposure on sleep/wake profiles in mice.

## 4. Discussion

This study revealed the alterations in sleep/wake patterns in mice following acute or chronic CORT exposure, echoing the sleep disturbances seen in humans subjected to acute or chronic stressors. The stress response constitutes multiple coordinated and dynamic processes, involving complex interactions between neural and hormonal networks, which might contribute to the varied and occasionally contradictory sleep changes ensuing from different stress types [12,14,16]. We aimed to dissect the mechanisms of stress-induced sleep disruption and concurrent behavioral abnormalities by focusing on the individual effects of the main glucocorticoid in rodent, which were previously difficult to discern under normal stress conditions.

To the best of our knowledge, this is the first study to analyze the effects of chronic CORT exposure on sleep architecture over a period of up to 28 days and a 7-day post-drug withdrawal stage. We discovered that the mice exhibited excessive wakefulness, depressive phenotypes, and elevated CORT levels during the 28-day administration. These changes were highly correlated and showed a progressive significance over the administration course. Notably, most of these changes were maintained up to 7 days after the termination of CORT exposure, which is a phenomenon rarely documented in rodent depression paradigms.

The acute surge In CORT levels precipitated a rapid promotion of wakefulness and concomitant suppression of both REM and NREM sleep, which is consistent with clinical observations wherein cortisol infusion markedly inhibited REM sleep [42], and with the rodent research which found that a psychological stressor (cage exchange) that caused acute CORT elevation led to reduced REM and NREM sleep during the second hour after the stressor [13]. There was no significant reduction in REM or NREM sleep beyond the fifth hour after injection, but there was a rebound in NREM sleep during the dark phase, presumably due to mounting homeostatic pressure due to previous sleep loss. The EEG delta activity is considered a biomarker of homeostatic sleep drive [43]. The increased EEG delta power density during wakefulness points towards a higher sleep drive consequent to the suppressed NREM sleep during the 4 h timeframe. In contrast, NREM sleep exhibited attenuated delta power, suggesting a subsequent decrease in sleep depth and attenuated sleep quality with CORT elevation, consistent with the decrease in slow-wave power observed in mice subjected to the foot-shock stimulation as an acute stressor [44].

Previous research on anxiety or depression has employed repeated CORT injections as the route of administration [45]. To avoid possible hazards associated with such injections [36,37] disturbing sleep/wake behaviour, CORT injection was only used during the acute exposure procedure, while chronic CORT exposure was achieved through a CORT-containing drinking water protocol. Another benefit of administration via drinking water is that stable doses of CORT can be more easily produced—a practicality that is challenging to achieve with injections in suspension.

The plasma CORT levels were raised remarkably along with robust depressive phenotypes after 28 days of exogenous CORT administration, which matched the studies of increased cortisol levels in depressive [46] and chronically insomniac patients [47]. The half-life of CORT in mice is about 20–30 min [48], so the elevated plasma CORT levels tested on D29 and D35, which was 1 day and 7 days after the drug withdrawal, could have many potential causes, including the increased energy demand accompanied with increased wakefulness. As CORT is the primary end product of the hypothalamic–pituitary–adrenal (HPA) axis in rodents, it is also possible that this elevation resulted from abnormal HPA axis hyperfunction. The hyperactivity of the HPA axis has been proved to be associated with depression in humans [49]. The enhancement could also arise from decreased CORT metabolism, which was reported in humans exposed to chronic stress [17]. Nevertheless, the elevated CORT levels may point to reduced sensitivity to glucocorticoid feedback in pathological conditions.

During this study, both the EPM test and the grooming bouts counting failed to find a significant difference between the CORT-exposed group and the control group. Hence, it is inadequate to determine whether long-term CORT exposure induces anxiety-like behaviour in mice. The NSF test has been noted for its sensitivity to chronic rather than acute antidepressant treatment, which is a feature distinct from other paradigms [50], thus reflecting its delayed manifestation within the context of depression. This distinctive trait may account for the maximal significance observed on D35 in the NSF test, 7 days following the conclusion of CORT administration, in striking contrast to other behaviour testing results.

The differential attenuation of NREM sleep on D28 by chronic CORT compared to D1 and D14, alongside the persistence of this decrease for a week post-withdrawal, underscores a robust correlation between elevated plasma glucocorticoid levels and chronic insomnia, as supported by clinical findings [51]. Other clinical evidence includes chronic insomnia in actively depressive patients [52] and patients with a history of depression [53]. Additionally, the absence of discernible alterations in each sleep/wake stage on D1 seemingly conflicts with the results from the acute CORT group. This divergence largely stems from the gradual elevation of CORT levels administered via drinking water, differing from the acute injection-induced rapid surge.

Changes in REM sleep occurred more rapidly, indicating its heightened sensitivity to CORT exposure. However, the increase in REM sleep on D14 and D28 were confined to the light phase, rather than the complete 24 h cycle, in stark contrast to wakefulness/NREM sleep alterations. The less pronounced alterations in REM sleep can be attributed to the overall reduction in sleep amounts. Notably, the significant increase in REM ratio from D14 to D35 correlates with clinical findings where an increase in REM sleep amounts and percent were reported in individuals with depression [54].

The EEG theta oscillations are the dominant rhythm in the hippocampus of rodents, and theta power increases with increasing task demands [55]. The decreased EEG delta power and increased theta power during the wake state after chronic CORT exposure indicated a higher vigilant and attentive state in mice. Interestingly, a clinical study had similar observations: waking EEG power in the theta band alongside plasma cortisol levels was increased in sleep-deprived participants [56]. We have noted that specific types of stress ostensibly leading to elevated CORT levels have been shown to induce sleep enhancement [14,15]. The possible explanation is that, sometimes, alertness is augmented to better cope with challenging environments, similarly to our findings, though sleep could be enhanced to relieve the malign effects of stress. By contrast, chronic CORT exposure progressively reduced theta activity during REM sleep, consistent with decreased theta activity observed in rats exhibiting increased REM sleep subjected to chronic stress [57]. The attenuated REM sleep theta oscillations may reflect some cumulative damage to the hippocampus along the administration course, which made sense since atrophy in the hippocampus is well-documented in depressive patients [58] and chronic CORT-treated mice [59]. Furthermore, decreased EEG delta power during NREM sleep following chronic CORT exposure matches the observations of lower EEG delta power in insomnia patients [60] and reduced slow-wave sleep quality in depressive patients. Noticeably, delta activity is known to contribute to cortisol suppression [61]. Therefore, the attenuated delta activity during the wake and NREM sleep stages might potentiate CORT elevation and the emergence of depression phenotypes.

The mechanisms through which CORT exposure sculpts neural circuits and finally elicits changes in sleep architecture remain obscure. To shed light on this matter, we performed c-Fos immunofluorescence staining to explore the impact of acute and chronic CORT treatment on different brain regions. As a critical node that interconnects forebrain and midbrain, the LHb orchestrates essential physiological functions, including sleep regulation. Studies have shown that aversive events led to an instant phasic excitation in LHb neurons and repeated exposure to stressors elicited hyperactivity of the LHb accompanied by depression-like phenotypes in animal models [62]. The PVH, upstream of the HPA axis, governs stress functions and modulates CORT output in rodents. It was reported to play a crucial role in the promotion and maintenance of wakefulness, with PVH dysfunctions leading to excessive sleep [63]. The VTA has been intensively investigated for its functions in goal- and reward-directed behaviour, but it has also been implicated in sleep-wake regulation. The activity and plasticity of its dopamine neurons could be influenced by CORT or acute/chronic stressors [64,65]. The LH, containing wake-promoting hypocretin/orexin (Hcrt) neurons, has been shown to display robust activity in response to stress exposure [12]. The BLA, integral in emotion and motivational processing, emerges as sensitive to stress and stress hormone CORT. Consistent with these previous studies, our c-Fos immunostaining results showed that acute or chronic CORT treatment promoted c-Fos expression in the LHb, coinciding with the emergence of depressive-like behaviour in the chronic paradigm. The role of LHb in the regulation of sleep/wake cycles remains poorly defined, with limited evidence from lesion studies [62]. However, our findings tentatively suggest that LHb may play a role in promoting wakefulness during stress-responsive scenarios. The activity of the PVH can be suppressed by an acute surge in CORT levels through negative feedback, which is consistent with our c-Fos findings. Conversely, following chronic CORT exposure, the PVH showed increased c-Fos immunoreactivity, suggesting a desensitized feedback mechanism akin to the impact of prolonged stress [17]. Considering the wake-promoting function of this nucleus [63], it is plausible that the PVH is implicated in chronic, but not acute, CORT-induced insomnia. Both acute and chronic CORT were shown to promote c-Fos expression in the VTA, validating its involvement in wake-promoting functions, potentially through activation of glutamatergic or dopaminergic neurons [66], although further co-staining identification is required. It is worth noting that both the PVH and VTA project to the LH, and acute CORT exposure, inducing rapid wake-promotion, increased c-Fos expression in the LH. Chronic CORT exposure also led to LH activation, albeit to a lesser degree. Collectively, these findings lend support to the role of the LH in mediating stress-induced heightened arousal. The hyperactivity of the BLA during prolonged stress conditions aligns with the observed results, wherein chronic CORT treatments prompted heightened c-Fos expression in the BLA. It should be noted that acute CORT failed to elicit a significant increase in c-Fos expression, indicating reduced responsiveness of the BLA to shorter periods of heightened glucocorticoid levels. Nevertheless, the inherent stress associated with acute treatment procedures cannot be neglected. It is plausible that the c-Fos expression in the control group was already elevated compared to the normal baseline, making it more challenging to detect an increase following acute CORT exposure. The prolonged exposure to CORT may have induced a diminished glutamatergic projections from the prefrontal cortex (PFC) to GABAergic interneurons in the BLA, leading to the loss of feedforward inhibition and subsequent hyperexcitability of BLA principal neurons [67,68]. Further application of optogenetic and chemogenetic techniques might offer a more insightful investigation into the neural circuitry mechanisms, through which CORT or stress processing impact sleep patterns.

## 5. Conclusions

This study substantiated that a single dose and 4-week regimen of CORT administration, with the latter inducing depressive-like behaviour and elevating CORT levels in mice, promoted wakefulness marked by increase episodes, and dampened NREM sleep characterized by augmented fragmentation levels and reduced EEG delta activity. Changes in sleep/wake parameters observed herein mirror the sleep disturbances that occur in patients with depression. Notably, the modifications in sleep and wakefulness persisted even 7 days subsequent to the cessation of CORT treatment. These findings warrant further investigation on the mechanisms through which chronically elevated glucocorticoid levels impact sleep within an animal model of depression.

## Figures and Tables

**Figure 1 brainsci-13-01472-f001:**
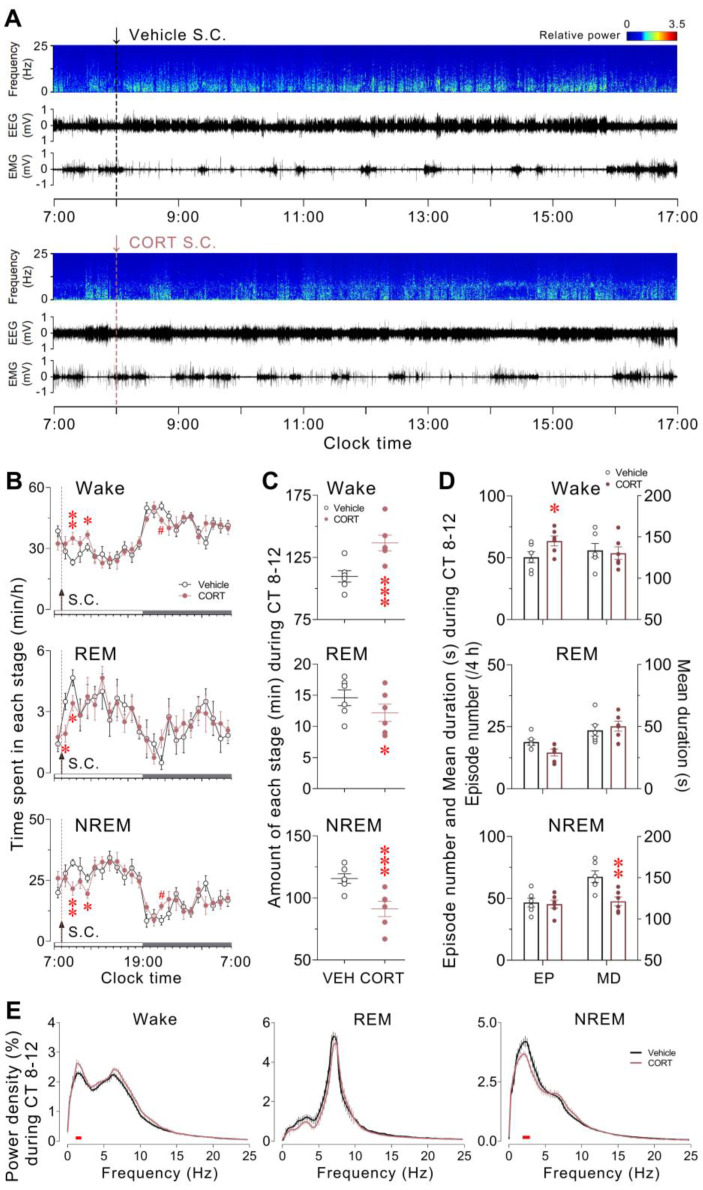
Acute CORT administration induced a rapid increase in wakefulness and a decrease in REM and NREM sleep accompanied by the altered EEG delta power. Mice received vehicle or CORT injection at ZT 1 (8:00 a.m.). (**A**) Typical examples of relative EEG power and EEG/EMG traces over a 10 h light period. The dashed line represents the vehicle (black) or CORT (red) injection. (**B**) Time course changes in wakefulness, REM sleep, and NREM sleep over a 24 h period following vehicle or CORT injection, marked by dashed lines. Data are subdivided into 60 min bins. The white and black bars above the *x*-axes indicate the light and dark phases, respectively. (**C**) Total time spent in each stage, (**D**) Episode number and mean duration of each stage and (**E**) EEG power density of each stage during the 4 h time window after vehicle or CORT injection. Values are presented as means ± SEM. (**B**) *n* = 6; two-way ANOVA (ZT 1 to 5), F_1, 10_ = 5.778 (Wake), 5.853 (REM), 5.402 (NREM), *p* = 0.0371 (Wake), 0.0361 (REM), 0.0425 (NREM), Bonferroni post hoc test, * *p* < 0.05, ** *p* < 0.01; two-way ANOVA (ZT 14 to 16), F_1, 10_ = 5.546 (Wake), 1.147 (REM), 5.533 (NREM), *p* = 0.0403 (Wake), 0.3094 (REM), 0.0405 (NREM), Bonferroni post hoc test, # *p* < 0.05; (**C**,**D**) *n* = 6; paired *t*-test, * *p* < 0.05, ** *p* < 0.01, *** *p* < 0.001; (**E**) *n* = 6; paired *t*-test, *p* < 0.05 is indicated by the red horizontal lines. CORT, corticosterone; CT, clock time; EP, episode number; MD, mean duration; NREM, non-rapid eye movement; REM, rapid eye movement; S.C., subcutaneous injection; VEH, vehicle.

**Figure 2 brainsci-13-01472-f002:**
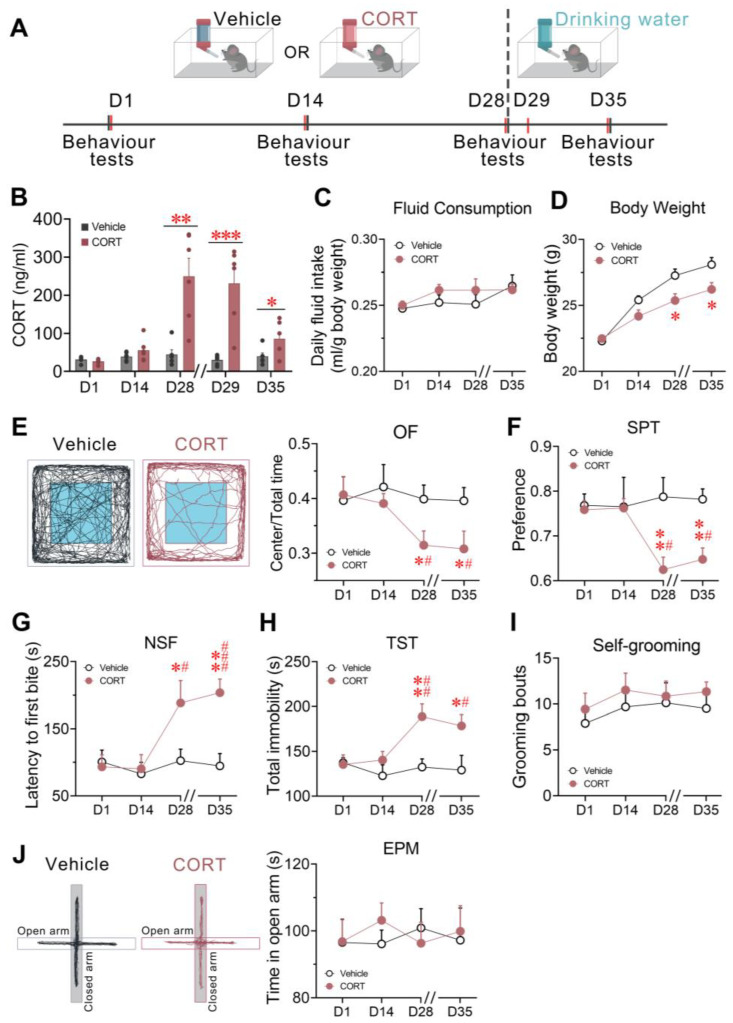
Chronic CORT administration elevated CORT levels and induced depressive-like behaviour in mice. (**A**) Schematic of the experimental procedure. Mice received 4-week administration of either vehicle or CORT via drinking water (D1 to D28), followed by a 1-week drug withdrawal period (D28 to D35). The short grey and red bars represent the timepoints of behaviour tests and plasma CORT determination, respectively. (**B**) Time course changes in plasma CORT levels across chronic administration and post-drug withdrawal period. (**C**) Fluid consumption and (**D**) Body weight monitored across chronic administration and post-drug withdrawal period. (**E**) Tracing of locomotion of representative animals on D28 (black: vehicle group; red: CORT group) and the ratio of time spent in the center zone to the total area during the OF test. (**F**) The ratio of sucrose water intake relative to total sucrose and regular water intake during the SPT. (**G**) The latency to the first food bite during the NSF test. (**H**) Duration of immobility during the TST. (**I**) Count of spontaneous grooming bouts within the 30 min observation window. (**J**) Tracing of locomotion of representative animals on D28 (black: vehicle group; red: CORT group) and time spent in the open arm during the EPM test. All double slashes on the *x*-axes signify the drug withdrawal. Values are presented as means ± SEM. (**B**) *n* = 6 per group; unpaired *t*-test, * *p* < 0.05, ** *p* < 0.01, *** *p* < 0.001; (**C**) *n* = 6 per group; (**D**–**J**) *n* = 9 in vehicle group, *n* = 12 in CORT group; (**C**–**J**) two-way ANOVA, F_1, 10_ =1.196 (fluid consumption), F_1, 19_ = 11.19 (body weight), 6.367 (OF), 9.861 (SPT), 10.76 (NSF), 11.72 (TST), 1.361 (Self-grooming), 0.0515 (EPM), *p* = 0.3240 (fluid consumption), 0.0034 (body weight), 0.0207 (OF), 0.0054 (SPT), 0.0039 (NSF), 0.0028 (TST), 0.2578 (Self-grooming), 0.8229 (EPM), Bonferroni post hoc test, * *p* < 0.05, ** *p* < 0.01 compared to vehicle group; paired *t*-test, # *p* < 0.05, ## *p* < 0.01, ### *p* < 0.001 compared to CORT group on D1. CORT, corticosterone; EPM, elevated plus maze; NSF, novelty suppressed feeding; OF, open field; SPT, sucrose preference test; TST, tail suspension test.

**Figure 3 brainsci-13-01472-f003:**
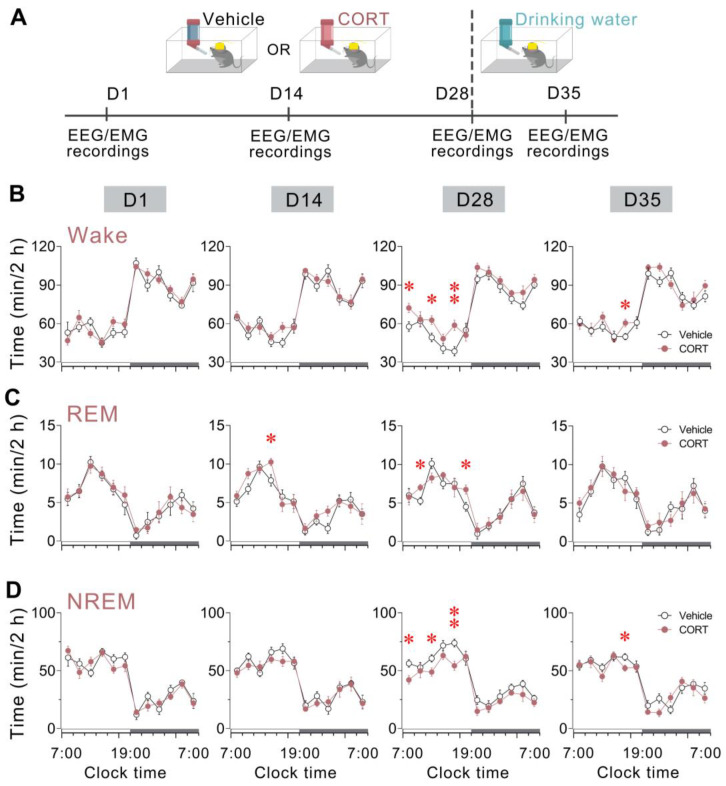
Chronic CORT administration promoted wakefulness and REM sleep while it suppressed NREM sleep. (**A**) Schematic of the experimental procedure. Mice received 4-week administration of either vehicle or CORT via drinking water (D1 to D28), followed by a 1-week drug withdrawal period (D28 to D35). The short grey bars represent the days of sleep-wake behaviour recordings. (**B**–**D**) Time course changes in (**B**) wakefulness, (**C**) REM sleep, and (**D**) NREM sleep across chronic administration and post-drug withdrawal period. Data are subdivided into 120 min bins. The white and black bars above the *x*-axes indicate the light and dark phases, respectively. Values are presented as means ± SEM. (**B**) *n* = 6 per group; two-way ANOVA (ZT 0 to 12), F_1, 10_ = 0.1378 (D1), 0.9854 (D14), 59.19 (D28), 7.178 (D35), *p* = 0.7233 (D1), 0.3592 (D14), 0.0003 (D28), 0.0366 (D35), Bonferroni post hoc test, * *p* < 0.05, ** *p* < 0.01; (**C**) *n* = 6 per group; two-way ANOVA (ZT 0 to 12), F_1, 10_ = 0.4000 (D1), 6.875 (D14), 9.524 (D28), 1.028 (D35), *p* = 0.5504 (D1), 0.0395 (D14), 0.0215 (D28), 0.3498 (D35), Bonferroni post hoc test, * *p* < 0.05; (**D**) *n* = 6 per group; two-way ANOVA (ZT 0 to 12), F_1, 10_ = 0.2250 (D1), 1.576 (D14), 58.40 (D28), 10.74 (D35), *p* = 0.6520 (D1), 0.2560 (D14), 0.0003 (D28), 0.0169 (D35), Bonferroni post hoc test, * *p* < 0.05, ** *p* < 0.01. CORT, corticosterone; NREM, non-rapid eye movement; REM, rapid eye movement.

**Figure 4 brainsci-13-01472-f004:**
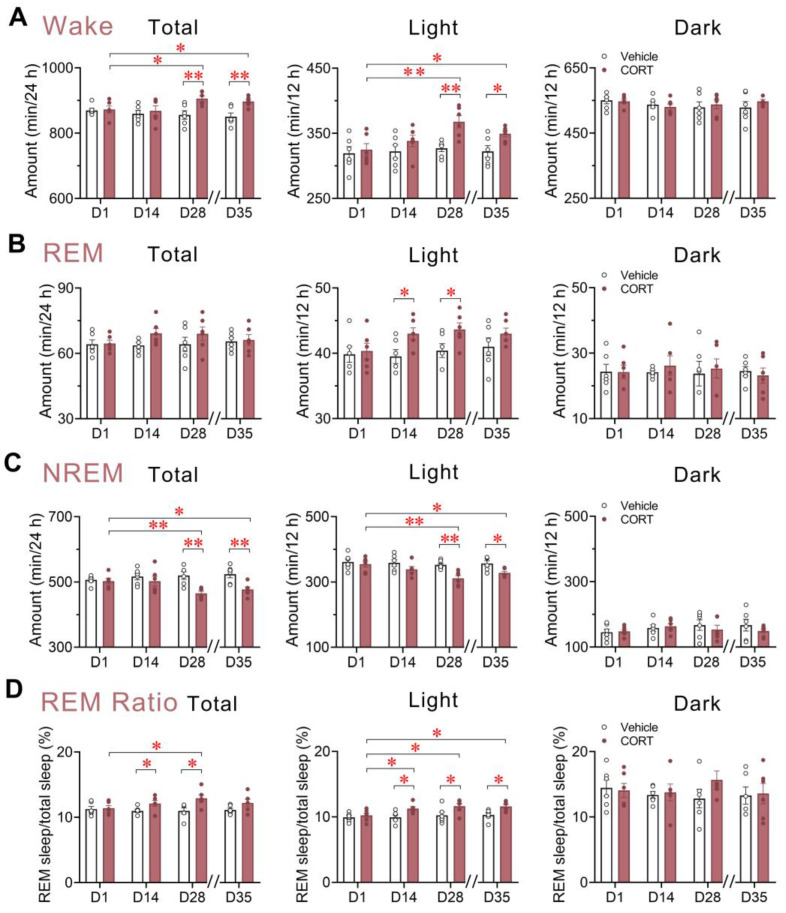
Changes in wakefulness, REM sleep, NREM sleep, and REM sleep ratio induced by chronic CORT exposure predominantly occurred during the light phase. (**A**–**D**) Total time spent in (**A**) wakefulness, (**B**) REM sleep, (**C**) NREM sleep, and (**D**) Ratio of REM sleep amounts to total sleep amounts during 24 h, light, and dark phases across chronic administration and post-drug withdrawal period. All double slashes on the *x*-axes signify the drug withdrawal. Values are presented as means ± SEM. (**A**–**D**) *n* = 6 per group; unpaired *t*-test, * *p* < 0.05, ** *p* < 0.01 compared to the vehicle group; paired *t*-test, * *p* < 0.05, ** *p* < 0.01 compared to the CORT group on D1. CORT, corticosterone; NREM, non-rapid eye movement; REM, rapid eye movement.

**Figure 5 brainsci-13-01472-f005:**
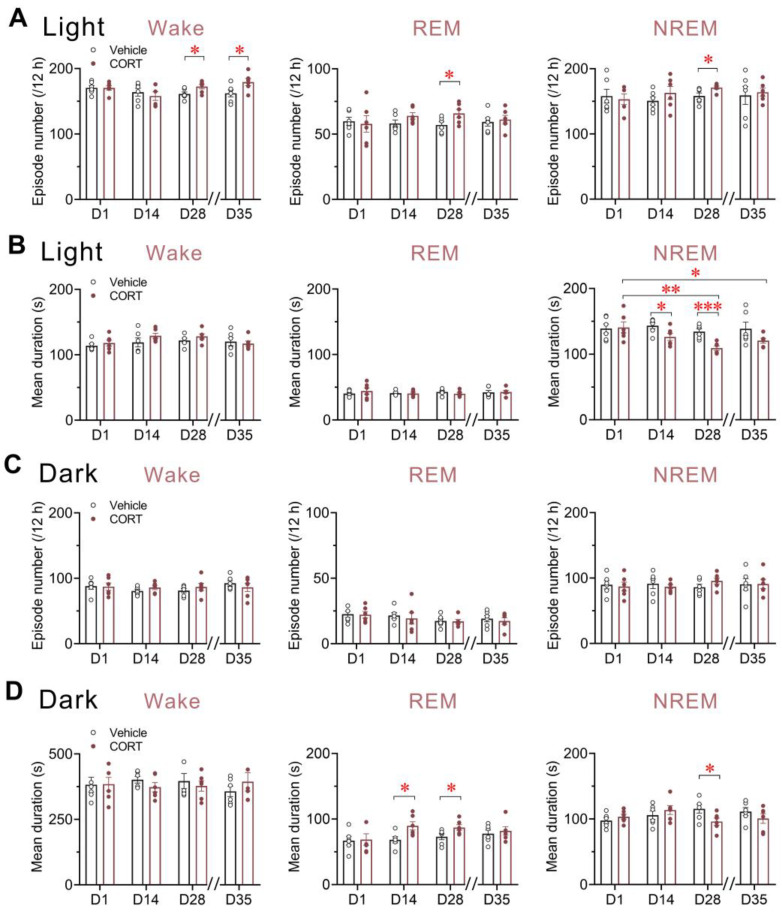
Chronic CORT administration increased episodes of wakefulness, REM sleep, and NREM sleep, prolonged REM sleep duration, and shortened NREM sleep duration. (**A**) Episode number and (**B**) Mean duration of each sleep/wake stage during the light phase across chronic administration and post-drug withdrawal period. (**C**) Episode number and (**D**) Mean duration of each sleep/wake stage during the dark phase across chronic administration and post-drug withdrawal period. All double slashes on the *x*-axes signify the drug withdrawal. Values are presented as means ± SEM. (**A**–**D**) *n* = 6 per group; unpaired *t*-test, * *p* < 0.05, *** *p* < 0.001 compared to the vehicle group; paired *t*-test, * *p* < 0.05, ** *p* < 0.01 compared to the CORT group on D1. CORT, corticosterone; NREM, non-rapid eye movement; REM, rapid eye movement.

**Figure 6 brainsci-13-01472-f006:**
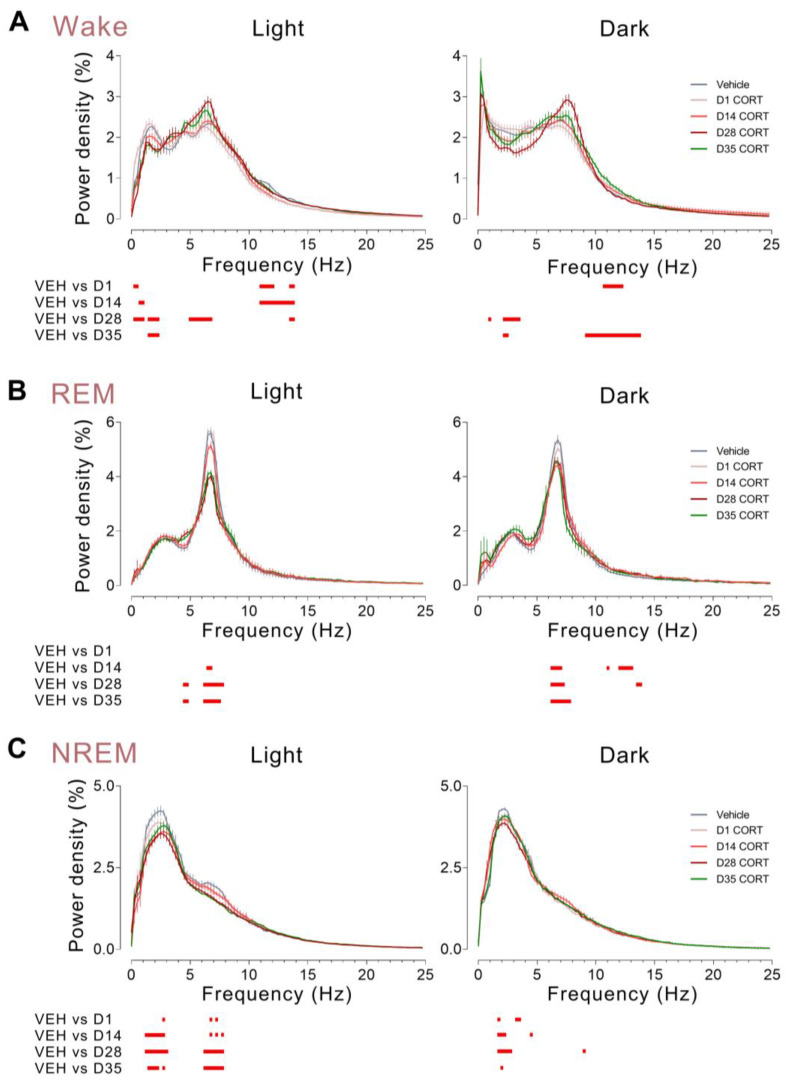
Chronic CORT administration decreased EEG delta activity and increased theta activity during wakefulness, while it decreased EEG theta activity and delta activity during REM and NREM sleep, respectively. (**A**–**C**) EEG power density of (**A**) wakefulness, (**B**) REM sleep, and (**C**) NREM sleep during light and dark phases across chronic administration and post-drug withdrawal period. Values are presented as means ± SEM. (**A**–**C**) *n* = 6 per group; unpaired *t*-test, *p* < 0.05 is indicated by the red horizontal lines. CORT, corticosterone; NREM, non-rapid eye movement; REM, rapid eye movement; VEH, vehicle.

**Figure 7 brainsci-13-01472-f007:**
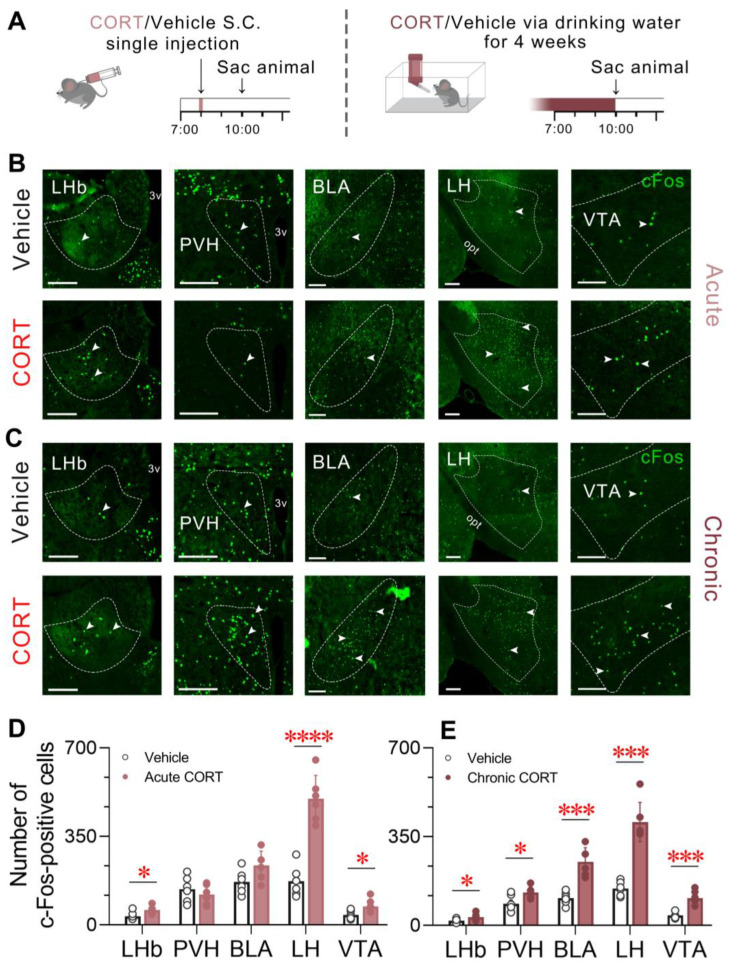
Effects of acute and chronic CORT administration on c-Fos protein expression in certain brain regions. (**A**) Schematic of the experimental procedure. Mice received either an acute CORT/vehicle injection (at ZT 1) or underwent chronic CORT/vehicle administration via drinking water for 4 weeks, followed by sacrifice at ZT 3. (**B**,**C**) Representative immunostaining images showing c-Fos-positive cells (green, marked by white arrowheads) in five distinct brain regions following (**B**) acute or (**C**) chronic CORT exposure, and the vehicle control. Scale bar, 200 μm. (**D**,**E**) Number of c-Fos-positive cells in five brain regions following (**D**) acute or (**E**) chronic CORT exposure, and the vehicle control. Values are presented as means ± SEM. *n* = 6 per group; unpaired *t*-test, * *p* < 0.05, *** *p* < 0.001, **** *p* < 0.0001. 3v, third ventricle; BLA, basolateral amygdala; CORT, corticosterone; LH, lateral hypothalamus; LHb, lateral habenula; opt, optic tract; PVH, paraventricular nucleus of the hypothalamus; Sac, sacrifice; S.C., subcutaneous injection; VTA, ventral tegmental area.

## Data Availability

The data and analyses used in this study can be obtained from the corresponding author with a reasonable request.

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
