# Peer review of "Acute or Chronic Exposure to Corticosterone Promotes Wakefulness in Mice"

_brainsci, 2023, doi:10.3390/brainsci13101472_

Round 1
Reviewer 1 Report
The manuscript by Yao and colleagues investigated the effects of acute and chronic corticosterone (CORT) on sleep-wakefulness and behaviors. The EEG data investigating the amount of wakefulness, REM and NREM sleep over a 24h period following acute and chronic CORT administration is particularly interesting. However, I do have a number of concerns regarding the experimental design and how it may impact interpretation of some of the data.
1. The acute dose of CORT of 20mg/kg seems extreme. Do the authors know how high circulating levels of CORT reach after s.c. administration? This is a dose that is commonly used to induce neural damage but is supraphysiological. The authors should justification why they chose this dose and how it may impact their results.
2. Similarly, it would be important to know how 0.035mg/ml CORT in the drinking water affects circadian levels of circulating corticosterone. It would also be important to monitor how much animals drink during the light versus dark phase. If chronic CORT administration results in increased wakefulness during the light phase as the authors report, do the animals also drink more during the light phase? Both increased wakefulness and increased ingestion of CORT water may explain some of the differences in circulating CORT the authors report.
3. I would caution the authors about directly comparing the cFos results from their Acute and Chronic study. The study designs are quite different. The acute study requires animals to receive injections 2h prior to perfusion. Injections are inherently stressful thus cFos expression in controls will be higher than normal baseline, this makes it easier to observe a suppression in cFos following acute CORT and more difficult to detect an increase. This is very different from chronic CORT administration in drink water where presumably the mice were left alone prior to anesthetizing animals for perfusion.
4. It is unclear why the authors express cFos as a ratio of the change from “baseline”. There is no true “baseline” as these are between subject data thus the variability in two separate groups needs to be considered.
5. The authors report that “Plasma CORT levels displayed no significant alternations during the initial phase of chronic exposure, remaining within the normal physiological range.” The authors should be more specific as they only took blood samples at one timepoint. The fact that aberrant CORT levels at ZT7-8 were detected after 28 days of CORT administration is interesting, but the authors should use caution in interpreting the elevated CORT as dysregulation of the HPA axis or impaired negative feedback (which were not tested here). The authors report chronic CORT administration results in increased wakefulness during the light phase which by itself would increase circulating CORT due to increased energy demand.
6. Was body weight recorded of the animals. Chronic CORT administration often results in reduced body weight gain across time. By day 28 there may be significant body weight difference between groups that could contribute to food/hunger drive in the NSF test.
7. The authors should provide more details about the order and timing between the behavioral tests, assuming the same set of animals were exposed to multiple tests. The authors should also clarify how many different cohorts of animals were used in the completion of the studies presented in the manuscript and what each cohort was used for. For example, did the animals used for cardiac blood collection also undergo behavioral testing or surgery for EEG recording and so forth?
Reviewer 2 Report
An interesting study of the acute and chronic corticosterone (CORT) administration on behavior and sleep in adult male mice. While the acute injection of CORT increased wake activity and reduced NREM sleep, the chronic 28-day exposure to CORT-containing drinking water induced depression-like behavior and still stronger sleep alterations, some of the effects being detectable even 7 days after exposure. The study contained careful analyses of sleep stages and EEG, the results of which were suggested to reflect an animal model for depression. There are still several points to be clarified before the analyses and the model can be fully understood.
1. Although a lot behavioral and other measures are reported, the report requires body weights, food and fluid consumptions to see how stressful the chronic procedure was, especially as there was a delayed dramatic surge of CORT plasma concentrations. Was there initially an aversion to drink the CORT-containing water?
2. The c-fos expression is very sensitive to injections and handling, to which the mice were apparently not being familiarized before the experiments. Therefore, it is difficult to see the value of comparison of the acute vs. chronic CORT as the acute stress effects could not be controlled for, especially as all the brain regions reported as stress sensitive ones. Please, either repeat the acute part or just present the data for both acute and chronic sets separately with addition of brain regions such as striatum and cerebral cortex, to counterbalance the selection bias as there was no prehypothesis on the brain regions. This analysis might also change the Discussion of the c-fos results, as the acute stress cannot be omitted using this present kind of procedure.
3. In the Introduction, please, justify the time of acute treatments as its fitting to the physiological CORT diurnal rhythm in mice, if it does. And how it affects the diurnal sleep rhythm.
4. The proportion of the delta rhythm is altered by CORT in acute and chronic experiments, which does not seem to be in strict relation to sleep stages, unless the automatic analyses missed NREM sleep in WAKE condition.
5. In Fig 5, please, use similar scaling of y-axes in all panels, to make it easier to compare light and dark phases for wake, rem and nrem.
6. In the Discussion, the last para on page 16 is difficult as it tries to explain something by hyperactivity of the HPA- axis, which should be normal in these mice, although might be abnormal in depressed human subjects? I suggest to change the structure of the discussion so that all comparison to human depression would be in one paragraph instead of being discussed in several paras.
Reviewer 3 Report
In this manuscript, Authors investigated how sleep was affected by acute or chronic CORT administration using EEG/electromyography (EMG) recordings in freely moving mice. According to this study, CORT exposure enhanced wakefulness, suppressed and fragmented NREM sleep, and altered EEG activity in all stages. This is a well-written manuscript, the author made a great effort, but I suggest some changes that must be made to the text, to contribute to a better understanding of the points they are trying to make.
comments
1. In line 33, what is EEG, please write the full form as it is used first time in the manuscript. In abstract author write some abbreviations, write all, or remove all. Please be consistent.
2. In line 80, material and method. Please explain the grouping of animals, like the number of mice in each group or in this study.
3. In line 46 and 47, please correct the line, Recent researches has suggested the intricate associations between stress and sleep.
4. In line 22, please correct the line, write were instead of was. For instance, single injections of CORT in rats was found to produce an initial enhancement of wakefulness and a reduction in slow-wave sleep during the first hour,
5. Authors must work on their English language as I have noticed many grammatical mistakes as well as some lines are tough to understand.
1.
2. In line 667 and 671, the author used the same reference. “Baglioni C, Battagliese G, Feige B, et al. Insomnia as a predictor of depression: A meta-analytic evaluation of longitudinal 667 epidemiological studies. J Affect Disorders 2011; 135(1-3): 10-9”.
Authors must work on their English language as I have noticed many grammatical mistakes as well as some lines are tough to understand.
Round 2
Reviewer 1 Report
The authors did a good job making appropriate revisions to address my concerns.
Reviewer 3 Report
Author made a great effort, Overall, the manuscript is improved. I am accepting this manuscript in present form.